# Diffusion MRI anisotropy in the cerebral cortex is determined by unmyelinated tissue features

Colin Reveley [1,2] ✉, Frank Q. Ye [3], Rogier B. Mars [1,4], Denis Matrov[5], Yogita Chudasama[5] & David A. Leopold [3,6]

Diffusion magnetic resonance imaging (dMRI) is commonly used to assess the tissue and cellular substructure of the human brain. In the white matter, myelinated axons are the principal neural elements that shape dMRI through the restriction of water diffusion; however, in the gray matter the relative contributions of myelinated axons and other tissue features to dMRI are poorly understood. Here we investigate the determinants of diffusion in the cerebral cortex. Specifically, we ask whether myelinated axons significantly shape dMRI fractional anisotropy (dMRI-FA), a measure commonly used to characterize tissue properties in humans. We compared ultra-high resolution ex vivo dMRI data from the brain of a marmoset monkey with both myelin- and Nissl-stained histological sections obtained from the same brain after scanning. We found that the dMRI-FA did not match the spatial distribution of myelin in the gray matter. Instead dMRI-FA was more closely related to the anisotropy of stained tissue features, most prominently those revealed by Nissl staining and to a lesser extent those revealed by myelin staining. Our results suggest that unmyelinated neurites such as large caliber apical dendrites are the primary features shaping dMRI measures in the cerebral cortex.

Recent advances in high-resolution diffusion MRI (dMRI) methods have revealed significant diffusion anisotropy in the cerebral cortex gray matter[1–8]. This finding provides an opportunity to extend the use of dMRI beyond its well-established application for characterizing white matter projections. Determining the cellular and histological features that give rise to this anisotropy is important, because it could offer a way for MRI methods to detect and measure the detailed cytoarchitectural properties of gray matter, both in healthy tissue and in disease states. Cytoarchitecturally grounded MRI contrasts could aid in the detection of neuropathology by linking foundational histological studies in the medical field[9] to in vivo studies in humans at the

level of individual subjects. For example, linking dMRI variation to tissue structure could be applied to track the progression of cortical pathology during the early stages of Alzheimer's disease[10], or to track tissue changes in conditions like PTSD[11].

The histological basis of diffusion anisotropy in gray matter is not yet well explored[12]. Based on principles derived from the analysis of white matter, where the majority of previous histological comparisons have been carried out[13,14], one possibility is that cortical dMRI patterns most directly reflect the density and arrangement of myelinated fibers[2,8,15–21]. Large-caliber axons, similar to those in white matter, can also be found in gray matter. For example some large incoming and

[1]Wellcome Centre for Integrative Neuroimaging, Centre for fMRI of the Brain (FMRIB), Nuffield Department of Clinical Neurosciences, John Radcliffe Hospital, University of Oxford, Headington, Oxford OX9 3DU, UK. [2]Department of Informatics, University of Sussex, Falmer, Brighton BN1 9QJ, UK. [3]Neurophysiology Imaging Facility, National Institute of Mental Health, National Institute of Neurological Disorders and Stroke, National Eye Institute, National Institutes of Health, Bethesda, MD, USA. [4]Donders Institute for Brain, Cognition and Behaviour, Radboud University Nijmegen, Nijmegen, The Netherlands. [5]Section on Behavioral Neuroscience, National Institute of Mental Health, National Institutes of Health, Bethesda, MD, USA. [6]Section on Cognitive Neurophysiology and Imaging, Laboratory of Neuropsychology, National Institute of Mental Health, National Institutes of Health, Bethesda, MD, USA. ✉e-mail: reveley@gmail.com

outgoing axonal projections run vertically through the cortex into the underlying white matter [8,22,23]. However, these are relatively sparse, and are only one aspect of gray matter myeloarchitecture. Myelinated axons in the gray matter are often organized very differently than in the white matter, with less dense packing, smaller caliber fibers, and sometimes a matted, rather than bundled, mesoscopic structure[24].

The spatial distribution of cortical myelin density varies substantially by horizontal location (anatomical region) and by vertical (laminar) position. The myelin content of voxels may be assessed using a wide variety of MRI contrasts[13,14,25], such as the T1w/T2w or magnetization transfer ratio (MTR). In fact, myelin density is currently the principal means by which MRI is used to estimate the histological boundaries of cortical areas in the human brain in vivo[26]. Quantitative assessment of MRI sensitivity to myelin, including the potential role of diffusion MRI (dMRI), is a topic of active research[13,14,25]. For dMRI specifically, some evidence suggests that the cortical variation of fractional anisotropy is determined more directly by the organizational structure of myelinated fibers, rather than by their density per se[3,27], although this prospect has not yet been demonstrated quantitatively[2]. Histological analysis of the cortex has revealed significant variation in the organization of myelinated axons from region to region,[24,28] to which diffusion anisotropy may be sensitive.

However, compared to white matter, myelinated axons are a comparatively small component of gray matter tissue structure. The cortical neuropil is dominated by unmyelinated substructures that may significantly shape the diffusion MRI signal, possibly outweighing or otherwise complementing the contribution of myelinated axons[29–31]. Further, it is known from previous work that myelin does not necessarily drive the compartmentalization of water molecules underlying dMRI anisotropy, even in myelinated axons, with lipid cell membranes being at least as important[31]. In the white matter, these elements covary because the overwhelming majority of neurites are myelinated axons, so the distinction between myelinated and unmyelinated neurites has not been of great importance[32]. However, given the greater tissue complexity of gray matter, recent theoretical and empirical studies have begun to explicitly consider the contribution of unmyelinated tissue components[2,3,33].

The most conspicuous anatomical feature of the cerebral cortex is its laminar structure. Previous dMRI studies investigating the cortex have consistently found that the principal diffusion direction is perpendicular (henceforth, "vertical") to the pial surface, with the exception of thin zones directly adjacent to the pial and white matter boundaries[3,34–36]. The diffusion orientations in white matter reflect the organization of axonal projections into directed bundles. The predominance of vertical diffusion in gray matter may reflect the organization of cell bodies, axons, dendrites and other tissue components into columnar structures[2,3]. Vertically oriented tissue features in the gray matter include some myelinated axons[22,23], but they also include, for example, apical dendrite bundles projecting vertically from deep to superficial layers[3,33,37–39] as well as the neurites of many other more regional and layer specific cell types[23,37,40].

In this study, we compared the spatial variation of dMRI anisotropy in the primate cerebral cortex to a range of histological features revealed through common tissue stains. We performed high-resolution ex vivo MRI scans of the brain of a common marmoset (Callithrix jacchus) and subsequently sectioned, stained, and co-registered the tissue, allowing for a pointwise comparison of dMRI fractional anisotropy (dMRI-FA) and tissue features across the gray matter. We found that the spatial variation of dMRI-FA showed little correspondence to that of myelin content, but that myelin content was instead well captured by a different MRI measure, namely the magnetization transfer ratio (MTR). However, the spatial variation of dMRI-FA corresponded well to that of structure tensor[41] derived histological anisotropy (HA) computed from the same sections, and particularly to that produced from Nissl staining. These findings bring to bear the

important role of unmyelinated tissue components, such as the prominent apical dendrites of infragranular pyramidal neurons[37–39], in shaping diffusion MRI signals in the cortical gray matter.

## Results

To assess the histological features underlying dMRI-FA in the primate cerebral cortex, we compared ex vivo high-resolution dMRI data from the brain of a common marmoset, to histological sections obtained from the same brain following the scanning session. The brain was scanned for 58 h on a 7 T Bruker preclinical scanner with 450 mT/m gradient coils. Six whole-brain volumes at 150 μm isotropic resolution were averaged for each of 60 diffusion encoded directions and two B0 volumes. This extensive averaging resulted in a dMRI dataset with a signal-to-noise ratio of ~85 in gray matter. In addition, magnetization transfer ratio (MTR) volumes were acquired in a separate session from the same brain at an isotropic spatial resolution of 75 μm. The MTR sequence is known to be sensitive to the myelin content of brain tissue[42]. The dMRI volumes were resampled to the higher MTR resolution for analysis using a spline interpolation. After MRI scanning, the brain's left and right hemispheres were sectioned into coronal and sagittal sections, respectively, at 50 μm on a freezing microtome. Adjacent sections were stained for Nissl bodies using the Thionin stain, and for myelin using the Gallyas silver stain. For quantitative analysis, nine strips of gray matter in the dorsal frontal and parietal lobes were chosen for comparison based on their low curvature, since cortical curvature can affect observed dMRI-FA values[43]. Nonlinear registration was subsequently applied, allowing for a direct and precise pixel to pixel comparison of diffusion and histological parameters at 150 μm spatial scale (see Methods, Supplementary Fig. 1).

### dMRI fractional anisotropy minimally reflects myelin content

We began our analysis with a simple pixelwise correlation of MRI and histological stain intensity within the nine registered sections. Our first question was whether the patterns of dMRI-FA matched the distribution of myelin intensity, as might be expected based on some previous studies in white matter regions[13,14]. We found that the spatial variation of dMRI-FA had poor correspondence to myelin intensity both across laminae (henceforth, vertically) and along the cortical sheet (henceforth, horizontally), as illustrated in a sagittal example section in Fig. 1a. Comparing the dMRI-FA to the myelin stain intensity revealed areas of apparent matching (e.g., blue arrow) and others of clear discrepancy (e.g., red arrow). In the vertical direction, myelin intensity declined monotonically from the white matter to the pia[44]. This differed from the laminar pattern of dMRI-FA, which peaked in the middle cortical layers (Fig. 1b)[4,45]. In contrast to dMRI-FA, the MTR MRI contrast closely matched the laminar pattern of myelin staining, as expected from the known sensitivity of the MTR MRI sequence to large macromolecules[42]. The myelin intensity showed relatively poor correspondence to dMRI-FA contrast and high correspondence to the MTR MRI contrast across all sections (Fig. 1c, Supplementary Fig. 2). We also found poor correspondence between dMRI-FA and Nissl stain intensity across all sections (Supplementary Fig. 2).

### dMRI-FA reflects anisotropy of cortical tissue features

Having established that myelin stain intensity was only modestly correlated to dMRI-FA, we next turned our attention to histological features most directly related to anisotropy. We generated parameter maps from the myelin and Nissl-stained sections based on the structure tensor (ST) model (see Methods, Fig. 2, Supplementary Fig. 3). Like the MRI-based diffusion tensor, the structure tensor derived from histological images offers both vector and scalar information about the tissue, in the form of principal orientations and ST coherence[46]. The ST calculation was weighted over a mesoscopic spatial area that matched the dMRI acquisition resolution of 150 μm (see Methods). Like the dMRI data, the spatial variation of the

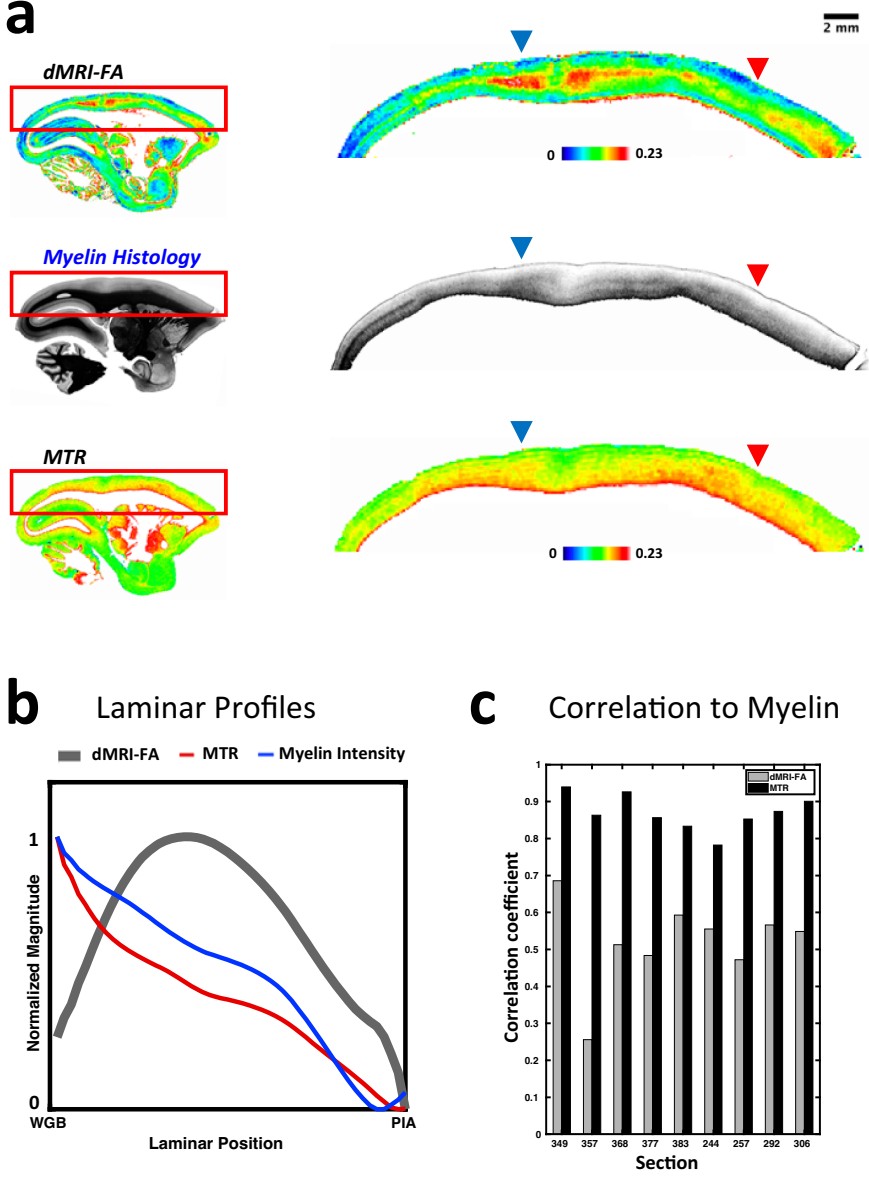

**Fig. 1 | Comparison of dMRI-FA to corresponding myelin histology and MTR MRI data obtained from the same tissue in the right hemisphere of marmoset monkey case P. a** Sagittal sections of dMRI-FA (top), Myelin histology (center), and MTR MRI (bottom). The spatial variation of dMRI-FA in vertical (laminar) and horizontal direction showed minimal correspondence with the pattern of myelin intensity, but the variation of the MTR MRI closely mirrored the myelin variation. In some regions of high myelination (dark colour in histology), the dMRI-FA was high (red arrow) whereas in others it was low (blue arrow). **b** Normalized average laminar profiles of FA, MTR and myelin intensity through the cortical thickness from roughly 100 μm above the white gray boundary (WGB) to roughly 100 μm below the pial surface (PIA) in the brain's left hemisphere, derived from cortical parcellation (see Methods, Fig. 3). **c** Histogram of correlation coefficients between MRI variables and histological myelin intensity over the nine sections of coronal histology used in this study. MTR MRI shows much higher correspondence with myelin intensity than dMRI-FA does. Source data are provided as a source data file.

ST parameters was resampled to 75 μm for analysis. For both the myelin- and Nissl-stained sections, the principal orientation of the ST was strikingly uniform and nearly always perpendicular to the pial surface (i.e., vertical), corresponding well to the orientation map obtained from the dMRI diffusion tensor (Fig. 2a). While it was not the focus of the present study, we also found that the 2D tracto-graphy computed from the histological ST and dMRI tensor data yielded qualitatively nearly identical results (Supplementary Fig. 4). Only in thin laminar zones directly adjacent to the pia and the white matter did this principal orientation deviate significantly from the vertical and differ across measurements.

The histological ST coherence is similar to dMRI-FA in that it reflects the degree of alignment of objects. However, unlike dMRI-FA[2,16], the magnitude of ST coherence does not incorporate the density of oriented features within an area[46] (Supplementary Fig. 3). We therefore devised measures of histological anisotropy (HA) that were more directly analogous to dMRI-FA by multiplying the histological ST coherence values with a pixelwise approximation of neurite density. To derive HA from the myelin stain (**myelin-HA**), we multiplied the myelin-based ST coherence with the normalized histological myelin density. For Nissl data, we computed HA (**Nissl-HA**) as the product of Nissl ST coherence and the Nissl derived "neurite fraction"[47] (see Methods). These measures served as summary statistics whose variation could be compared systematically with that of the dMRI-FA across the gray matter (Fig. 2b). While the inclusion of neurite density into the HA measure was deemed appropriate for this comparison,

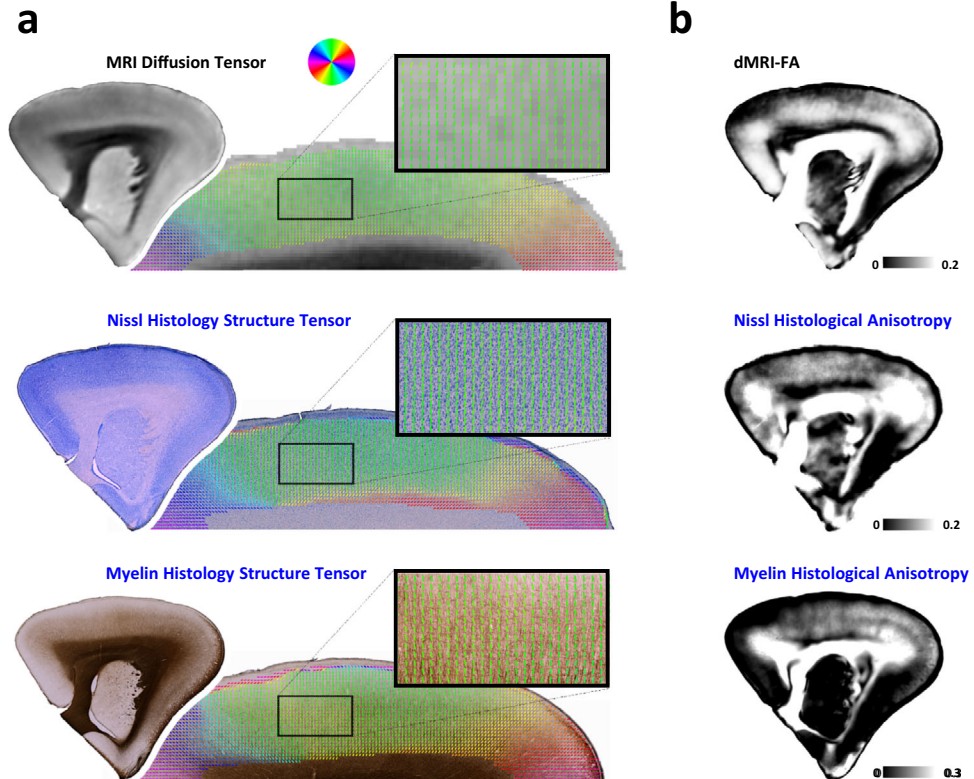

**Fig. 2 | Properties of directional MRI and histology data.** For comparison with MRI data, the weighted structure tensor of the myelin and Nissl histology images was computed over a 150 μm area (see Methods). **a** Orientation data derived from Nissl and myelin histology and from dMRI are similar, showing columnar, vertically oriented vectors. The overall pattern of orientations in the gray matter is remarkably similar for the data computed from histology and from dMRI. **b** The right column shows dMRI-FA and nonlinearly registered histology anisotropy maps which show the degree of edge dispersion from the mean in 75 μm tiles (see Methods).

qualitatively similar results for all figures were obtained using maps of the histological coherence alone (Supplementary Fig. 5).

For each section, we quantitatively compared the spatial variation of dMRI-FA to the Nissl-HA, myelin-HA, and myelin stain intensity (MI) scalar parameter maps (see Methods). For completeness, we also compared the MTR MRI data to the same histological parameters. To accommodate anatomical variability and potential inconsistencies in histological stain intensity, we employed a parcellation approach in our analysis of the gray matter. In each of the nine low-curvature sections, we compared the spatial variation in the MRI and histological parameter maps by computing the correlation over multiple parcellated subdivisions spanning the thickness of the cortex (e.g., red trapezoids in Fig. 3, see Methods, Supplementary Fig. 6). We focused analysis on a horizontal parcellation width of 300 μm along the pial boundary, though the results of the analysis were robust to parcel sizes ranging over a hundred-fold in area (see Supplementary Fig. 7). Within a given parcel, a pixel-by-pixel correlation of variation within each column revealed different levels of correspondence for the different histological and MRI parameters. This difference can be seen in the example parcel in Fig. 3. For this parcel, the dMRI-FA showed the strongest spatial correspondence to the histological anisotropy of the Nissl-stained sections and somewhat less to that of the myelin-stained section (Fig. 3a, bottom). It did not correlate well with the profile of myelin intensity (MI), which had a rather different laminar profile. However, the magnetization transfer ratio (MTR) MRI, exhibited a strong correlation with the myelin distribution (Fig. 3b, bottom), consistent with previous findings[13,14].

We evaluated such correlations across all 393 parcels in the nine sections for the different MRI and histological parameter maps, noting the anatomical region of each parcel with reference to the Paxinos et al

atlas[48,49]. The distributions of dMRI-FA correlations across all parcels to the three histological parameter maps are shown in Fig. 4a. The color of each point represents the cortical area from the atlas. Across the cortex, the Nissl histological anisotropy (Nissl-HA) showed consistently higher correlations with the dMRI-FA ($\rho_{median} = 0.76$, $\rho_{mean} = 0.72$) than did the myelin anisotropy ($\rho_{median} = 0.63$, $\rho_{mean} = 0.57$) or myelin intensity ($\rho_{median} = 0.47$, $\rho_{mean} = 0.39$). The spatial maps of myelin intensity closely matched those of the MTR MRI contrast (Fig. 4b), ($\rho_{median} = 0.94$, $\rho_{mean} = 0.93$). Nissl-HA and myelin-HA were correlated moderately to one another (Supplementary Fig. 7). For all histological parameters, parcels with relatively high and low tissue correlation coefficients were generally intermixed within each atlas area. One exception to this intermixing was the correlation of dMRI-FA with agranular regions such as primary motor cortex (4ab), which had notably high correlations. A similar pattern of results was obtained when we substituted an in-plane, two-dimensional measure of dMRI anisotropy[41] for the three-dimensional dMRI-FA measure (Supplementary Fig. 8). The closer correspondence of dMRI-FA to the Nissl- rather than the myelin-stained tissue features indicates that, across the cerebral cortex, unmyelinated tissue structures drive the spatial variation in high-resolution dMRI-FA maps.

Because fractional anisotropy can itself be broken down into diffusivity components, we further investigated how each of these components tracked our measured tissue features. Fractional anisotropy (dMRI-FA) is related to the ratio of axial diffusivity (AD) to radial diffusivity (RD), with the former corresponding to diffusion along the principal diffusion axis, and the latter corresponding to diffusion perpendicular to this axis. An increase in dMRI-FA can thus be effected either by a relative increase in AD (vertical diffusivity) or a decrease in RD, and we found dMRI-FA was linked to both in the gray matter

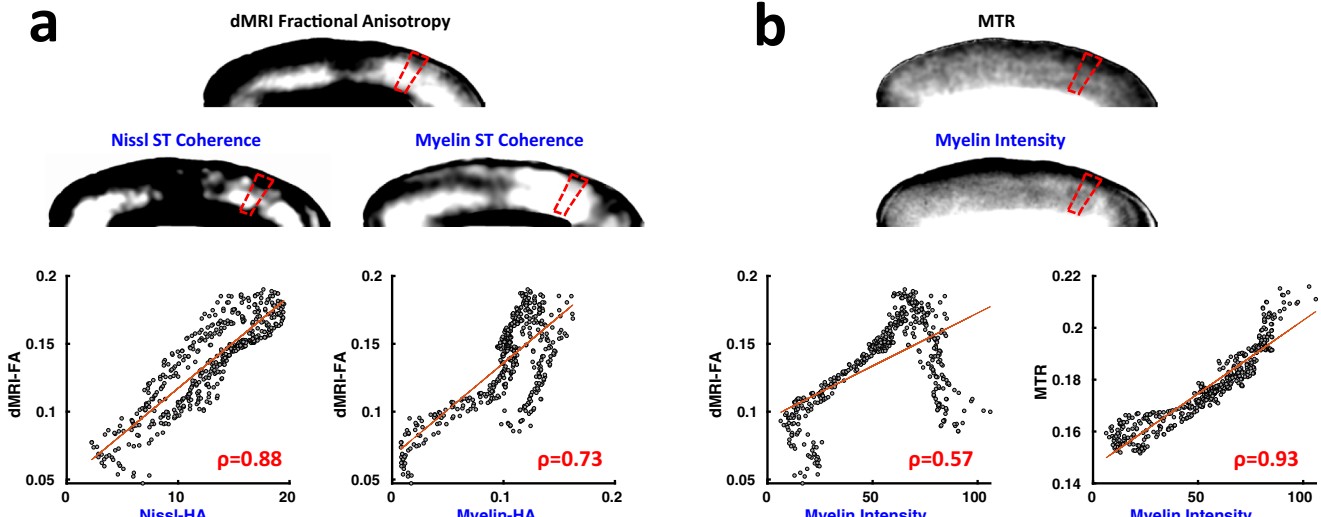

**Fig. 3 | Example correlation of Histology and MRI variables sampled in columnar parcels. a** Comparison of dMRI fractional anisotropy to corresponding Nissl-HA and Myelin-HA from one parcel of corresponding tissue. **b** Comparison of magnetic transfer ratio (MTR) MRI contrast with myelin intensity in same parcel as **a**. In both **a** and **b**, the top row depicts nonlinearly co-registered histology parameter maps and MRI at 75 µm resolution. The red trapezoids indicate a columnar parcel example. The bottom row shows representative pixelwise scatter plots of the co-registered data within the sample parcel of cortex, and Pearson correlation (ρ) coefficients between histological variables and MRI variables. ST structure tensor. Source data are provided as a source sata file.

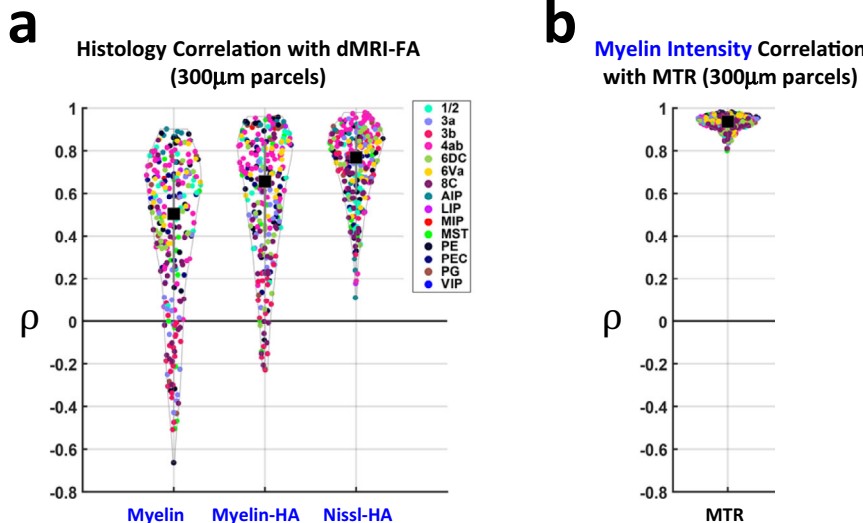

**Fig. 4 | The distribution of correlation coefficients between histological parameters and dMRI-FA overall 393 parcels of cortex. a** correlations of histology variables to dMRI-FA, Myelin: min −0.66, Q1 0.18, median 0.47, Q3 0.69, max 0.9; Myelin-HA: min −0.23, Q1 0.39, median 0.64, Q3 0.83, max 0.96; Nissl-HA: min 0.11, Q1 0.61, median 0.76, Q3 0.87, max 0.98. **b** Correlation of MTR to Myelin Intensity: min 0.8, Q1 0.91, median 0.94, Q3 0.96, max 0.98. Each point in the distribution is the Pearson correlation between the variables sampled in a vertical parcel (see Methods). Colors indicate the anatomical region a parcel falls within. Black square shows median value and vertical gray bars show lower and upper quartiles. ρ indicates Pearson correlation coefficient. Each parcel is one of $n = 393$ independent samples from the marmoset cortex, spanning 18 distinct histological sections. Source data are provided as a source data file.

(Supplementary Fig. 9). Previous work has established a strong association between lowered RD and myelin content in coherently organized white matter[19], reflecting the tendency of myelin to restrict diffusion perpendicular to axons. Consistent with these findings, we found that the spatial variation of RD was strongly anticorrelated with that of both myelin stain intensity and myelin-HA in the cortical gray matter (Supplementary Fig. 9, Supplementary Fig. 10, Supplementary Fig. 11). However, the spatial variation in axial diffusivity, quantifying diffusion in the vertical direction, had no discernable relationship to myelin-HA, and was moderately anticorrelated to myelin intensity. Nissl-HA was more modestly anticorrelated to RD, however unlike myelin-HA it was also positively correlated to AD. These findings

suggest that myelinated axons, though tightly linked with the variation of RD, have only a secondary contribution to the overall spatial distribution of dMRI anisotropy in the cerebral cortex.

### Horizontal variation in dMRI-FA is tracked by Nissl-HA, not myelin-HA

Given that the laminar (vertical) and tangential (horizontal) dimensions of the cortex differ substantially in their architectural variation, we performed additional analyses to consider these dimensions separately. First, we addressed the vertical dimension. We analyzed the mean laminar variation of dMRI-FA and compared it to the histological anisotropy (Fig. 5a). Collapsing the data across all horizontal cortical

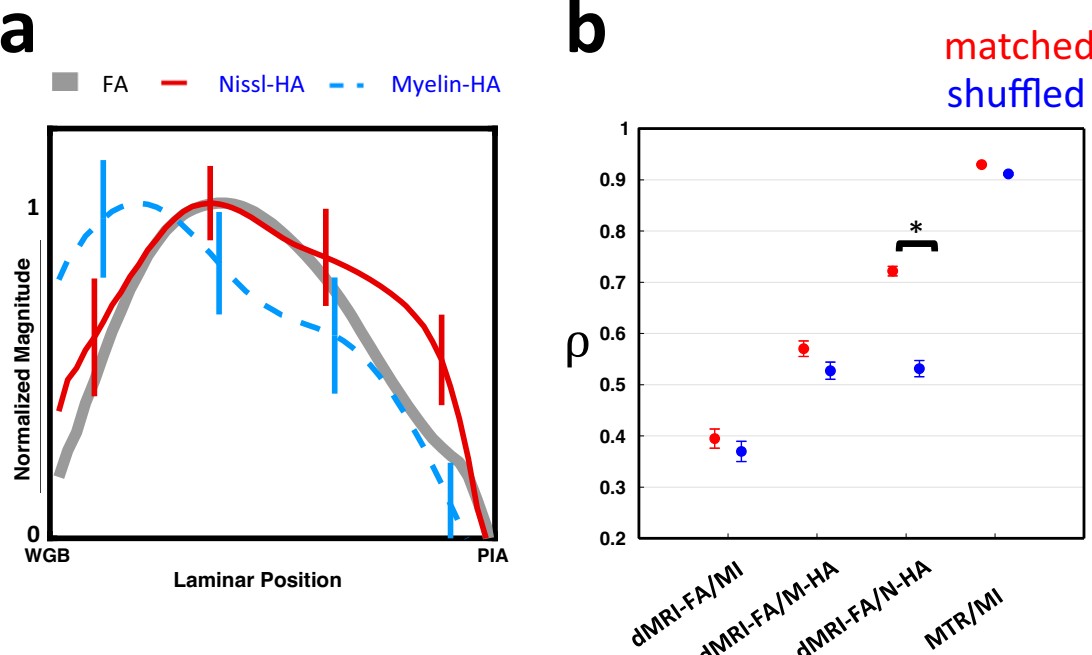

**Fig. 5 | Average laminar profiles of FA, Nissl-HA, and myelin-HA across all parcels and the results of a spatial randomization condition to test the regional specificity of parcel-wise histology to MRI correlations.** The results show Nissl-HA more closely matches the laminar profile of dMRI-FA, and is more sensitive to shuffling of horizontal MRI locations. **a** Normalized laminar profiles of dMRI-FA, Myelin-HA, and Nissl-HA through the cortical thickness, from roughly 100 µm above the white/gray matter boundary (WGB) to roughly 100 µm below the pial surface (PIA), computed from the average of all columnar parcels (see Methods, Fig. 3).

Bars indicate standard deviation. **b** The mean parcel correlation coefficients between MRI and histology variables from matched locations (see Fig. 4), and when the cortical parcels of the histology parameters and those of the dMRI-FA are shuffled between random horizontal locations. MI myelin intensity, N-HA Nissl-HA, M-HA myelin-HA. Bars indicate standard error. The star indicates significance under a two tailed paired $t$-test, $p < 10^{-10}$. ρ indicates mean Pearson correlation coefficient. Each parcel is one of $n = 393$ biologically independent samples from the marmoset cortex, spanning 18 distinct histological sections. Source data are provided as a source data file.

positions (see Methods) revealed a pronounced laminar profile for each of the measured parameters. The dMRI-FA profile had its highest responses in the middle or lower-middle cortical layers, with a decline in anisotropy in both upper layers, and lower layers near the white gray boundary (WGB). This profile was consistent with results from human dMRI[4], although our higher resolution data revealed significantly more underlying variation in the laminar profiles of individual horizontal locations (Supplementary Fig. 12, Supplementary Fig. 13). The average laminar profile of dMRI-FA across all horizontal regions in Fig. 5a resembled the observed histological anisotropy, but with stronger correspondence to the Nissl-HA measure. In particular, like dMRI-FA, Nissl-HA took a low value near the white gray boundary and peaked in middle layers, but myelin-HA had a relatively high value in the deepest layers. As described earlier, the profile of dMRI-FA differed markedly from the more monotonic profiles of myelin intensity and MTR (Fig. 1c). The average laminar Nissl-staining intensity exhibited nearly the inverse profile to the myelin intensity, having its highest intensity in the superficial layers (Supplementary Fig. 14).

Given these conspicuous average laminar profiles, we next asked whether the observed correspondence between dMRI-FA and the two measures of histological anisotropy could be ascribed to laminar variation alone, or also to variation in the horizontal direction, reflecting tissue differences between anatomically defined cortical areas. To address this question, we shuffled the horizontal locations of the columnar parcels in the histological parameter maps, relative to the locations of their corresponding dMRI-FA parcels. If such shuffling were to have no effect on the observed parcel correlations, then we could conclude that the results in Fig. 4 are fully explained by laminar variation that follows a stereotypical pattern across the whole cortex.

We found that horizontal shuffling strongly affected the correlations of Nissl-HA with dMRI-FA, but did not significantly change the

myelin-based histological/MRI correlations (Fig. 5b). For Nissl-HA only, horizontal shuffling caused the mean correlation to drop significantly, from 0.72 to 0.52 ($t$-test, $p < 10^{-10}$). These results indicate that the Nissl-HA measure has unique, region specific cytoarchitectural correspondences to dMRI-FA that go beyond the laminar profile of the cortex in general. In contrast, the relationships of myelin-HA and myelin intensity to dMRI-FA lack this regional specificity. In both those cases, and in the case of the relationship between myelin intensity and MTR, the average laminar profile in the cortex is sufficient to characterize the relationship between dMRI-FA and histology.

## Contribution of apical dendritic processes

The peak value of dMRI-FA is in middle layers, approximately from layer 5 to layer 3, which is congruent with Nissl-HA but differs from both myelin-HA and myelin density (see Fig. 5a, Supplementary Fig. 13). Nissl-HA serves as an indirect marker for unmyelinated tissue features. However, some evidence suggests that it reflects, more than any other tissue component, the apical dendrites of layer 5 pyramidal neurons. These dendrites are exceptionally large-caliber unmyelinated neurites that run vertically towards the pia, coalescing into bundles as they course through layer 4[22,37–39,50,51]. This universal feature of isocortex has been described in detail in the macaque[38,39], human[52], rat[37,39,50], and other species[39]. Figure 6 shows an example of this prominent structural feature of the cerebral cortex in a second marmoset subject, which we stained for the protein MAP-2 using immunohistochemical methods. MAP-2 (microtubule associated protein) is a structural protein that is abundant in the soma and dendrites of those pyramidal cells with the largest dendrites[38]. Figure 6a shows that the position and orientation of the pyramidal cell bodies reflect the orientation of their apical dendrites, and the overall vertical anatomy of cortex, with its bundles of large, unmyelinated dendrites[23,37–40].

The features revealed in the MAP-2 stained tissue (Fig. 6) were consistent with both the principal diffusion orientation (Fig. 2) and the peak diffusion anisotropy observed in the middle cortical layers (Fig. 5a). In layer 6, the deepest, multiform cortical layer adjacent to the white matter, Nissl-HA and dMRI-FA take lower values than myelin-HA (Fig. 5a). In this layer, pyramidal cell body orientations are reported in the literature as "tilted"[37,38] with dendrites emerging at a variety of angles. In Fig. 6b we confirm this incoherence in the neurites of layer 6 in the marmoset (Fig. 6b). It bears emphasis that this deepest portion of the cortex has the highest density of gray matter myelinated axons (Fig. 1a, b), and also high ST anisotropy of those axons, (Fig. 5a, Supplementary Fig. 13) yet these do not lead to elevated dMRI-FA in this location. In layer 5 and into layer 4 the MAP-2 labeled pyramids are larger, more plentiful and more vertically oriented. Their dendrites tend to bundle together[37–39] as they project upward (Fig. 6c). Dendrites of infragranular pyramidal cells have their tufts in layer 4 or deep layer 3[37–39], thus forming a lower tier[37–39] of apical dendrite bundles. An upper tier of dendrites takes its origin in the smaller, less coherently oriented pyramidal cells in Layers 2 and 3[37–39], reflected by the weaker MAP-2 labeling in the marmoset (Fig. 6d). Thus, while many tissue features may contribute to the restriction of water diffusion in the circuitry of the cerebral cortex, the apical dendrites of layer 5 pyramidal neurons are a ubiquitous cortical feature that appear to have a particularly important role.

## Discussion

The results of this study demonstrate that diffusion MRI in cortical gray matter can capture important, hitherto inaccessible cytoarchitectural tissue properties in vivo. We compared histological data to fractional anisotropy of the diffusion tensor (dMRI-FA), a widely used and straightforward dMRI parameter with minimal a priori assumptions about tissue properties[53]. Our point-by-point analysis (Fig. 3, Fig. 4) of exceptionally high-resolution, finely co-registered dMRI-FA and histological parameter maps from the same brain indicated that, in contrast to many findings in the white matter[13,14], myelin does not principally shape the mesoscale variation of diffusion anisotropy in the cerebral cortex.

On deeper investigation, we found that myelin density and histological anisotropy tended to take high values in the deepest cortical layers, while both dMRI-FA and Nissl histological anisotropy took high values in the central cortical layers (Fig. 1b, Fig. 5a). Importantly, we found that the pattern of dMRI-FA correlations with Nissl-HA was disrupted by shuffling the horizontal locations of the tissue parcels, but the correlations with myelin-based parameters were not (Fig. 5b). This suggests dMRI-FA is predominantly shaped by the regional cytoarchitectonic differences reflected in Nissl-HA, rather than by the myeloarchitecture. We found that these results are consistent with patterns found in apical dendrites, notably the vertically projecting, bundled apical dendrites of layer 5 neurons[23,37–40,50–52,54]. This is a universal feature of the cerebral cortex, and these dendrites may be strong contributors to the restriction of diffusion (Fig. 6).

The cerebral cortex is characterized by distinct anatomical zones, or regions, and the properties of these regions may be tracked by dMRI-FA. However, many distinct cortical maps have been proposed, based on different cytoarchitectural, myeloarchitectural, connectional and functional properties[55,56]. Our cortical parcellation strategy (Methods, Fig. 3, Supplementary Fig. 6) was designed to accommodate this variety, by systematically assessing the tissue over many differently sized subdivisions (Supplementary Fig. 7). Our results were robust to a range of parcel sizes, whose horizontal extents spanned two orders of magnitude. This tolerance to parcel size suggests the correlation results were minimally affected by changes in stain properties within a section, or by parcels that spanned cytoarchitectural boundaries. Similarly, the absence of an obvious correspondence between the observed parcel correlations, and any particular

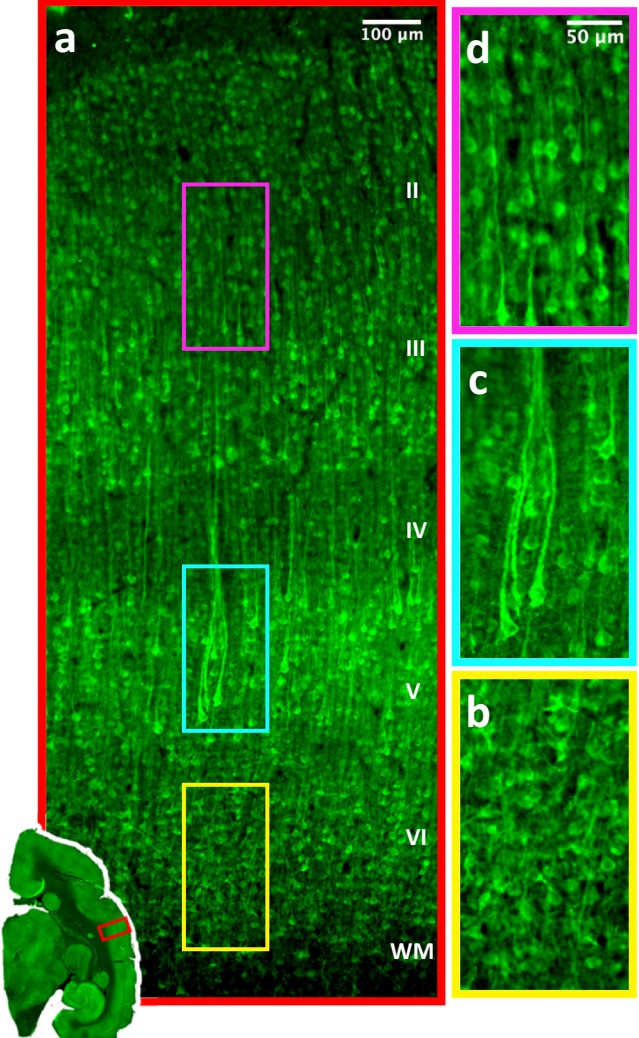

**Fig. 6 | MAP-2 immunohostochemistry labeled tissue data qualitatively matches the profiles of dMRI-FA and Nissl-HA. a** The stain reveals the larger apical dendrites and associated pyramidal cell bodies. **b** In layer 6 cell body orientations and dendrites are smaller and incoherently organized. **c** Layer 5 pyramidal cell bodies are larger, orientated vertically and project large vertical dendrites which bundle as they proceed into layer 4. **d** Cell bodies in layers 3 and 2 are smaller and apical dendrites less visble. MAP-2 staining carried out in one animal, these structures were representative of the n sections we stained. We examined 6 such sections which showed similar results.

cytoarchitectonic regions, indicates that the high point-by-point correlations between Nissl-HA and dMRI-FA we observed are a feature that generalizes over the cortex.

An exception to this general pattern was found in layer 4 of granular cortex, where there was sometimes a mismatch between low Nissl-HA, and high dMRI-FA (Supplementary Fig. 15a), leading to somewhat diminished correlations. The relatively lower Nissl-HA values in layer 4 likely stemmed from the predominance of round granular cell bodies. Given the vertical orientation in cortex, the high dMRI-FA values[4,5] in layer 4 most likely reflect the prominence of apical dendrite bundles coursing through the granular layer toward the pial surface[23,37–40,50–52,54], an organization that was strongly evident in our MAP-2 staining. Other neurites, such as layer IV terminals of thalamic afferent axons[57] (Supplementary Fig. 15b), and the unmyelinated, vertical axons of layer 4 cells themselves[23], likely contribute to the restricted diffusion in this layer as well. In areas of relatively low granularity, these apical dendrites are adequately indicated by

orientation cues in the Nissl stain. However, in highly granular areas, the Nissl-HA indicates diminished anisotropy in layer 4.

The specific association of Nissl-HA and dMRI-FA outside of layer 4 is likely related to the arrangement of the cell bodies into cellular minicolumns for which apical dendrite bundles are thought to provide a substructure[23,37–40,52,54]. Cellular minicolumns in Nissl-stained data have been used to detect and quantify the architecture of the cortex[23,58–60], and recent studies have begun to explore their relationships to parameters derived from the diffusion tensor in ex vivo human data[61,62]. Apical dendrite bundles exhibit a hexagonal packing structure when viewed in tangentially sectioned tissue[37,52,54]. In the vertical plane, soma of all cell types are packed vertically along the cells of this honeycomb-like structure[37,52,54], and these packed cell bodies exhibit composite vertical edges[52], which are likely to be detected by our structure tensor methods[23,37,38,40,63] (Supplementary Fig. 2). A similar structure tensor based method has very recently been applied to detect white matter fiber bundle orientations via arrangements of oligodendrocytes in Nissl data[64].

In addition to apical dendrites, dMRI-FA is likely also shaped by other unmyelinated tissue features, such as the axons of double bouquet cells[23,39] or layer IV terminals of thalamic afferent axons[57]. Immunohistochemical stains for proteins such as MAP τ[65], SMI-312[66], SMI-311[66], GFAP[67] and others may be useful to investigate the relative contributions of other non-myelinated tissue features, though quantification of neurite structure is difficult using antibody staining. For example, MAP-2 is a protein implicated in microtubule assembly, which is disproportionately expressed in the largest dendrites, thus leaving thinner dendrites unstained, even if they are numerous[37]. The specific combination of tissue features that shape cortical dMRI-FA cannot be fully known at this point and may well differ between anatomical regions. Further research is required to understand the columnar anatomy of the cortex itself[37,63], especially in a quantitative way. With that said, the strong spatial covariation (median 0.76) of Nissl-HA to dMRI-FA, and the regional specificity of that relationship, holds promise that the structural basis of diffusion MRI in the cortex can be well understood.

Our finding that Nissl-HA and myelin-HA correlated in distinct ways to the diffusivity components of dMRI-FA may offer some further clues to its anatomical basis. We found that the cortical variation of dMRI-FA, which is related to the ratio of axial to radial diffusivity, was as strongly driven by increases in its axial component as by decreases in its radial one (Supplementary Fig. 9). In the samples of gray matter we studied, AD is always diffusivity in the vertical direction, perpendicular to the pial surface. Of the histological measures, only Nissl-HA positively correlated with axial diffusivity. Myelin-HA did not have a consistent relationship to AD, however it did have a notably strong and negative relationship to radial diffusivity (Supplementary Fig. 9b, Supplementary Fig. 11), indicating its known role in restricting diffusion perpendicular to the orientation of axons[19,20]. However, the strong anti-correlation of RD and myelin-HA did not predict the spatial variation of dMRI-FA per se, which was strongly driven by variation in axial diffusivity. We do not yet know why Nissl-HA and dMRI-FA reliably covary with AD, while myelin parameters and MTR do not. However, unlike myelinated axons, apical dendrites have a reliably vertical orientation[23,37–40,52,54]. Combined with their large numbers, and potentially also with differences in intrinsic diffusivity between neurites[68], variations in apical dendrite density may be responsible for this effect.

The distinct diffusivity profiles of myelin-HA and Nissl-HA are a matter for further research, because they might be used in biophysically motivated models of the diffusion signal[12,68,69] that can distinguish between myelinated and non-myelinated tissue compartments. In dMRI models of white matter, it is sometimes assumed that the intrinsic rate of diffusion in the axoplasm parallel to the membrane is constant[68,70]. In the more heterogenous environment of gray matter, this assumption may not hold[12] and instead, the cytoplasm of different

cell compartments may have distinct intrinsic diffusivities[31]. Recent modeling efforts in white matter have sought to simultaneously measure variation in intra-axonal diffusivity, and variation in axon orientation[68]. Related biophysical models may prove especially important in gray matter. Models that include myelin levels, cortical curvature[61,62] and location in the cortical depth may be able to gather very detailed information about the human cortex in vivo.

Accurate gray matter diffusion models for applications related to human health must be grounded in decisive empirical data. The variation of diffusion MRI parameters may reflect valuable information such as cytoarchitectural differences between cortical regions, or degradation of tissue due to disease[61,62]. Measurements taken in vivo at low resolution will represent the influences not only of cyto and myeloarchitecture, but also of cortical curvature[43,61,62]. High-resolution data in ex vivo animal models is important for establishing the histological basis of signals that can be obtained in humans, now or in the near future as imaging capacities continue to improve. In our exceptionally high-resolution ex vivo dMRI data, we found the cortex was dominated by structures running perpendicular to the pial surface, and the simple diffusion tensor model was sufficient to capture the variation in anisotropy. However, when imaging the brain at lower resolution in vivo, more complex models may be necessary to represent the heterogeneous tissue properties contained within the much larger voxels. The marmoset brain has a cortical thickness and cytoarchitectural complexity that is similar to the human, and it can be used as a model of human neurodegenerative processes[71]. Moreover, its small brain volume and lissencephalic cortex allows for the study of histological dMRI correlates in detail and at high resolution, without the effects of curvature. This information may form a basis for the interpretation of signals from the highly gyrencephalic human brain scanned at lower resolution[61,62].

Diseases such as Alzheimer's are believed to involve subtle changes to tissue properties especially in their early stages, and the ability to gather specific cytoarchitectural information separately from myeloarchitectural in vivo may be crucial[10], particularly since relatively little post-mortem information is available regarding the early tissue changes. The early stages of Alzheimer's and other diseases of the gray matter neuropil can manifest as cellular damage and inflammation rather than demyelination[72]. Changes in mean diffusivity in gray matter have been associated with Alzheimer's pathology[73]. The present findings may inform the development of dMRI models of the gray matter, perhaps to specifically target neuroinflammation[10] or other aspects of Alzheimer's pathology in the earliest stages.

## Methods
We describe below the methods used in this study, sample preparation and MRI acquisition, sectioning and staining, registration and finally statistical analysis.

### Animals and materials
The brain of one adult marmoset (female, 5-year-old, "Case P") from a previous study[48] was used for MRI to histology comparison. Tissue from a second brain, (female, 2-year-old, "Case F"), was obtained from an unrelated study to examine the MAP-2 immunohistochemistry. All the procedures and provision of materials for this study were in full compliance with the Guidelines for the Care and Use of Laboratory Animals by National Institute of Health and approved by the Animal Care and Use Committee of the National Institute of Mental Health.

### Ex vivo high-resolution MRI acquisition
Before MRI scanning, the formalin-fixed marmoset brain of case P was soaked with 0.15% gadopentetate dimeglumine (Magnevist, Bayer Leverkusen, Germany) for 3 weeks to reduce the T1 relaxation time. In order to improve SNR, brain P was cut through the midline, and each hemisphere was scanned separately with a 25 mm birdcage volume

coil. MRI Data was collected on a 7 T Bruker preclinical scanner with 450mT/m gradients, and Bruker Paravision 5.1 software.

For the left hemisphere Diffusion MRI was collected with a 3D diffusion-weighted spin-echo EPI sequence with a spatial resolution of 150 µm isotropic voxel size. The 3D diffusion EPI sequence parameters were: TR = 400 ms, TE = 28.4 ms, flip angle = 90°, FOV = 38.4 × 24 × 21 mm, matrix size = 256 × 160 × 140, resolution = 0.15 mm isotropic. Sixty two DWI volumes (2 at b = 0, and 60 directions with b = 4800) were acquired using 3 averages. The same 62 volumes were collected a second time with identical settings. The total DWI acquisition time was about 58 h. The pulse width for the diffusion weighting gradient is 6.4 ms and pulse separation is 14 ms.

The right hemisphere was scanned with 100 µm resolution. The parameters for the 3D diffusion-weighted spin-echo EPI sequence were: TR = 400 ms, TE = 35 ms, flip angle = 90°, FOV = 38.4 × 20 × 25 mm, matrix size = 348 × 200 × 250, resolution = 0.10 mm isotropic, a total of 132 DWI images (6 at b = 0, and 126 directions with b = 4800), and total acquisition time was about 55 h.

The left and right hemispheres separately underwent scanning for MTR at 75 µm and 150 µm respectively.

The MTR images were collected with a 3D FLASH sequence. For the resolution of 150 µm, the parameters were: TR = 21.5 ms, TE = 3.6 ms, flip angle = 20°, FOV = 38.4 × 24 × 21 mm, matrix size = 256 × 160 × 140, number of averages = 8 and number of repetitions = 6, total acquisition time about 13 h For the resolution of 75 µm acquisition, the parameters are: TR = 21.9 ms, TE = 3.8 ms, flip angle = 20°, FOV = 34.8 × 20.4 × 20.1 mm, matrix size = 464 × 272 × 268, number of averages = 5 and number of repetition = 14, total acquisition time about 60 h. The magnetization transfer was achieved (in Msat scans) by a Gaussian shaped pulse at +2000 Hz offset frequency in one scan and at −2000 Hz in the other scan of a paired acquisition; the duration of pulse is 12.5 ms and flip angle is 540° and the pulse is present in every TR. This magnetization transfer pulse is turned off in $M_{off}$ scans. The MTR value is calculated as $100*(1 − M_{sat}/M_{off})$.

## MRI processing

**Diffusion tensor estimation.** dMRI data was preprocessed using TORTOISE v3[74] which includes code for denoising[75]. After rotation to the histology slice plane (see *Registration of histology and MRI data* below), the diffusion tensor was estimated using FSL 5.0.11 DTIFIT. AD, RD and the in-plane anisotropy[41] were calculated using FSL 5.0.11 fslmaths.

For estimation of cytoarchitectural boundaries, we registered the Paxinos et al. atlas[48] to the left hemisphere of case P MTR using ANTS 2.4.1[76].

Registered dMRI tensor parameter maps and MT data were split into separate coronal 32 bit quantitative tif images and grouped according to slice number for registration.

## Histological processing of MRI brain

Case P's left hemisphere was coronally sectioned at 50 µm on a freezing microtome, while the right hemisphere was sectioned sagittally. Alternate sections were stained for Nissl bodies and for myelin. The Gallyas silver stain for myelin was selected to stain individual axons in high detail[77]. We used the Thionin stain for Nissl bodies. This is a traditional staining method that identifies cell bodies and some dendrites proximal to the soma, but very few axons[78]. Each histological section was scanned at 0.88 µm resolution (×10 magnification) using a Zeiss Axioscan microscope slide scanner and onboard Zeiss Axioscan software. Higher resolution 0.44 µm microscopy produced similar ST results. Nine matched pairs of Thionin/Gallyas stained sections were selected for nonlinear registration based on criteria of even staining and minimal damage.

## MAP-2 immunohistochemistry

The brain of a second marmoset, case F, was used to gather MAP-2 immunohistochemistry. The brain's right hemisphere was sectioned at 40 µm on a freezing microtome. A free-floating brain section was submitted to the heat-induced epitope retrieval to enhance immunolabeling by incubating it for 30 min in 10 mM Tris-EDTA buffer (pH 9.5) at 80 °C[79]. Then the section was incubated for 60 h at 4 °C with a 1:200 dilution of a rabbit anti MAP-2 primary antibody (188 003, Synaptic Systems, Germany) and for 3 h with a 1:250 dilution of a goat anti-rabbit secondary antibody conjugated to DyLight 680 fluorophore (35569, Invitrogen, Waltham, Ma). Finally, the section was mounted on a glass slide, air-dried, and cover-slipped with DEPEX mounting media (13515, Electron Microscopy Sciences, Hatfield, PA).

## Registration of histology and MRI data

**Rigid body rotation to histology slice plane.** For histology to MRI registration, we first performed an initial rigid body alignment of the 75 µm $M_{sat}$ data (Supplementary Fig. 1) to match the histology slice plane for each of P's hemispheres. This transformation was estimated by-eye using the mango MRI editor/viewer[80] and the manually transformed MT volume was used as the target for FSL 5.0.11 FLIRT[81] to transform the remainder of the MRI data.

**Registration of MT and dMRI.** The 150 µm dMRI was rigid body registered to the 75 µm MT volume using FSL FLIRT, and in doing so the dMRI was upsampled to 75 µm using spline interpolation. The b-vectors were rotated using the FLIRT transform and custom MATLAB code.

**MIND nonlinear registration of local histology ROI.** After performing ST computations on the histology to produce ST anisotropy parameter maps (see Structure Tensor Analysis of Histological Orientations and Anisotropy below), we performed careful nonlinear alignment of nine local frontal and parietal ROI from the histology to the MRI (Supplementary Fig. 1). Stain intensity maps of nine coronal sections of each of Nissl and myelin histology at 75 µm resolution were first binarized using a threshold adjustment in imageJ[82] and then registered to their corresponding MRI slices to correct for histology shrinkage and deformation, first using manual 2D alignment in imageJ[82] and then MIND MATLAB code[83] (α = 0.1). To avoid registration between MRI and histological features whose anatomical meanings and relationships were unknown, the registration target images were featureless gray matter masks derived from the MT data (Supplementary Fig. 1). Custom MATLAB code was written to store and apply these transforms to ST maps (see Structure Tensor Analysis of Histological Orientations and Anisotropy below) and other parameter maps derived from the histology. We selected regions of frontal and parietal cortex for nonlinear alignment whose minimal gyrification ensured that the diffusion MRI signal was largely unaffected by cortical curvature[43]. This registration process allowed us to directly compare the cortical variation of diffusion and histological parameters from the same brain.

## Structure tensor analysis of histological orientations and anisotropy

To estimate the anisotropy and orientation content of the histology we employed the structure tensor method[41]. The 0.88 µm per pixel resolution histology images were processed using the structure tensor as implemented in the "OrientationJ" ImageJ package[46], using a local gaussian window of σ = 85 pixels and image gradients computed with a cubic spline. This was applied to the color channel of the histology that exhibited the strongest image intensity gradients. This was the red channel for Nissl data, or blue channel for Gallyas myelin data. Orientation and anisotropy information were obtained from these data

according to

$$S_0[p] = \begin{bmatrix} I_x[p]I_x[p] & I_x[p]I_y[p] \\ I_x[p]I_y[p] & I_y[p]I_y[p] \end{bmatrix} \quad (1)$$

Equation 1. Pixelwise structure tensor where $S_0$ is the structure tensor at pixel p and $I_x$, $I_y$ are the partial derivatives of the 2D image intensities at pixel p. Since the dMRI data contains diffusion information gathered over a 150 μm cubic area, the histology structure tensor at each pixel was a distance weighted function $S_w[p]$ of the structure tensors $S_0$ within a radius of 75 μm. This was achieved for pixel p by multiplying with gaussian window with a sum of 1, with sigma 85 pixels (75 μm), shown in Eq. 2. The ST angle is shown in Eq. 3 and anisotropy in Eq. 4 and Supplementary Fig. 15

$$S_w[p] = \sum_r w[r]S_0[p-r] \quad (2)$$

Equation 2. Weighted structure tensor

$$\theta[p] = \frac{1}{2}\arctan\left(2\frac{I_x[p]I_y[p]}{I_y[p]I_y[p] - I_x[p]I_x[p]}\right) \quad (3)$$

Equation 3. ST orientation

$$\text{SCoherence} = \frac{\lambda_1 - \lambda_2}{\lambda_1 + \lambda_2} \quad (4)$$

Equation 4. Structure tensor coherence ($\lambda$ indicate eigenvalues of the structure tensor).

A pipeline was implemented in bash and ImageJ macro language to process the Zeiss axioscan histology output files in CZI format into 32 bit tif format with resolution 0.88 μm, and process these with OrientationJ. The histology stain intensity data and ST data at 0.88 μm were then down sampled into maps of 75 μm tiles for comparison and registration to MRI data using MATLAB R2018a. The mean ST angle within 75 μm was the circular mean of the 0.88 μm angles within that area. Because ST anisotropy is sensitive to coherence of image edges, we verified that individual myelinated axons were visible on our 50 μm sections in the deepest, most myelin heavy layers of the GM (Supplementary Fig. 16). We also verified that ST coherence maps were equivalent at different focal depths in the tissue (Supplementary Fig. 17).

**Histological anisotropy.** Because diffusion anisotropy is sensitive to both the density and coherence of tissue features, but ST coherence is sensitive only to the latter, we defined composite measures of histological anisotropy. Myelin-HA was defined as the product of myelin stain intensity and the ST coherence of the myelin data.

For the Nissl-HA we defined the neurite fraction (NF) as the laminar inverse of the Nissl intensity, where the highest gray matter value was mapped to 1 and the lowest was mapped to 0. Nissl-HA was defined as the product of NF and the ST coherence of the Nissl data.

**2D dMRI and histology ST tractography**
The ST analysis of the myelin and Nissl histology yielded 2D vector fields at 75 μm resolution representing the mean 0.88 μm ST angle within a tile. For comparison, we projected the 3D dMRI principal eigenvector at the same spatial resolution onto the 2D plane (Fig. 2).

The vector fields of the 150 μm weighted structure tensor sampled at 75 μm resolution permitted tractography based on histological image properties within areas equivalent to the MRI voxels. We ran identical tractography through the Nissl and myelin ST vector fields. We used a MATLAB package "even streamlines" v 1.1.0.0 to ensure the streamlines were evenly spaced in the horizontal direction[84], modified

to use a fixed integration step of 0.1. Supplementary Figure 4 shows tractography through the 2D projected components of the first eigenvector of the diffusion tensor, overlaid on high-resolution histology images. We note all three modalities yield very similar results.

**Cortical parcellation and statistical analysis**
To sample the gray matter for analysis, we took advantage of the fact that cortex consists of vertically oriented tissue components[22,34,37,85] to develop a matrix representation whose columns were derived from data sampled along the vertical dMRI tractography streamlines (Supplementary Fig. 6a) (see 2D dMRI and Histology ST Tractography above), and whose rows spanned the horizontal direction. We first masked the registered gray matter ROI by thresholding the MT image ($M_{sat}$), and eroding this mask by 2 75 μm pixels to avoid the regions where the diffusion tensor orientations were known not to be vertical (see Fig. 2). We then ran tractography through the 2d projected principal eigenvectors of the diffusion tensor in the masked region. This produced a set of vertical, non-intersecting streamlines (Supplementary Fig. 6a) which were numbered from medial to lateral in the horizontal direction.

For each histology and MRI parameter map, we sampled a vector of unique pixels whose coordinates intersected one streamline's coordinates. We then resampled each resulting vector of pixels to a fixed length using bilinear interpolation, to yield a simple matrix representation of the gray matter (Supplementary Fig. 6a). This operation was invertible such that data in matrix form could be displayed in image space (Supplementary Fig. 6b).

Our statistical analysis proceeded by parcellating the matrix representation of the nonlinearly registered histology and MRI data. We used parcellation in our analysis partly to separate out regions of potentially differing stain intensity, and partly because different anatomical regions of cortex have different constituent myelo- and cytoarchitectural elements[22,38] and may therefore exhibit different relationships between histology and dMRI. For each parcel we took the Pearson correlation between sample variables. Examples are shown in Fig. 3. Supplementary Figure 6b shows parcels of matrix columns of different sizes, represented in image space and colored by the correlation coefficient of histology ST anisotropy to dMRI-FA.

Figure 4 and Fig. 6 show the distribution of correlation coefficients using a parcel size of 5 streamlines which is roughly 300 μm in horizontal width. Supplementary Figure 6 shows that the correlation results are stable for different parcel sizes. For Fig. 5 we concatenated matrices from all sections and displayed the mean of all rows. Supplementary Figure 9 shows the mean of rows in each individual section.

**Reporting summary**
Further information on research design is available in the Nature Research Reporting Summary linked to this article.

## Data availability
The source data generated in this study have been deposited in a Figshare database found at https://doi.org/10.6084/m9.figshare.21342473. Preprocessed data can be found at https://doi.org/10.6084/m9.figshare.21342341. Source data are provided with this paper.

## Code availability
The custom scripts used for data analysis in this study are available in a Figshare repository at https://doi.org/10.6084/m9.figshare.21342341.

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

## Acknowledgements

This research was supported by the Intramural Research Program of the National Institute of Mental Health (ZIC MH002899 to D.A.L. and ZIA MH002951 to Y.C.). C.R. and R.B.M. were supported by the Wellcome Centre for Integrative Neuroimaging (Wellcome Trust, 203139/Z/16/Z), and a Biotechnology and Biological Sciences Research Council grant to RBM (BBSRC UK,BB/N019814/1). Special thanks go to Dr. Afonso Silva for providing the MRI scanner time and instrument support. Scanning of histological brain sections was carried out in the Systems Neuroscience Imaging Resource under Dr. Ted Usdin in NIMH. We thank David Yu for histology processing.

## Author contributions

C.R. devised and initiated the study; C.R., F.Q.Y., R.B.M., and D.A.L. designed research; F.Q.Y. acquired the principle data; C.R. developed the analysis strategy, wrote the code, and analyzed the data; D.A.L. and

F.Q.Y. refined and discussed the analysis strategy; D.M. conducted the MAP-2 immunohistochemistry; C.R. wrote the manuscript; D.A.L., F.Q.Y., D.M., Y.C., and R.B.M. edited the manuscript; All authors discussed results.

## Competing interests

The authors declare no competing interests.
