## [Peer Review File · Nature Communications]

Diffusion MRI Anisotropy in the Cerebral Cortex is Determined by Unmyelinated Tissue FeaturesREVIEWER COMMENTS

Reviewer #1 (Remarks to the Author):

The paper by Reveley et al explore the cellular relevance of diffusion MRI measurements in the cortex.

Before providing my review of this paper, I would like to note that over the development of diffusion MRI as a tool in neuroscience it became fixated that this method is used for white matter mapping (mainly due to fiber-tracking and its consequent mapping of white matter structure in 3D). I strongly believe, as the paper points out, that diffusion in the cortex is highly meaningful and provide micro-structural view of cortical architecture. For that reason, and due to the popularity of diffusion MRI in neuroscience and it's unfortunate tagging as white matter mapping tool – I think this paper deserves high-impact publication.

The paper nicely demonstrates through direct comparison between one Marmoset brain MRI (diffusion and MTR) with various histological measures the relation between anisotropy measurements in the cortex and cellular morphology. Specifically, I think Figures 3 and 4 demonstrates that main result of this paper (mainly Fig. 3 dMRI-FA/Nissl-HA and MTR/Myelin intensity graphs; please consider adding A, B, C, D tagging to this figure) and those should be highlighted over other results.

I have only minor comments for the authors to consider:

1. Abstract: "Those unmyelinated features" needs better phrasing - Dendrites? Neural Processes? Unmyelinated Axons? (all of them?). Need to be more explicit.

2. Introduction, general – The authors state that diffusion MRI in the cortex is dominated by fiber arrangement perpendicular to pial surface. This is evidently true. However, in such a heterogeneous environment - does this perpendicular dominance really significant? In my view - only more generalized models than the diffusion tensor (e.g. spherical harmonics) can capture the complexity of organization in the cortex (some of the references the author cite also indicates that). Yet, the validity of the simple diffusion tensor model increases with extreme resolution where partial volume induced complexity reduces dramatically. It appears that this is the case in this work – 150micron resolution in the cortex should be sufficient so that the diffusion tensor model can be used. Maybe the authors would benefit from acknowledging that diffusion in the cortex necessitates more complicated models, but due to high resolution that is used, the simple diffusion tensor is adequate.

3. Introduction, last paragraph - Histological anisotropy - is that a common index of histology? A reference would be helpful. Later on in the results another terminology is used - structural tensor (ST) - this reminds me of a procedure by Budde that also explored correlation between histology and dMRI in the cortex. Is that a similar procedure? If so, a reference could be helpful.

After reading the methods section as well, I realized that indeed this is the case – so just reference Budde also in the main text.

4. 58 hours of scanning – Was the data corrected for signal drift? From my experience such long scanning may suffer from drift artifact which may have devastating effect on the diffusion anisotropy measurements.

5. What does the blue circles in the violin plot (Fig 4) indicates?

6. The Paper is long. The main message is already well received after Fig.4. The RD/AD correlation could be moved to the supplementary material. The layer 4 neuronal type analysis (Fig 6 and on) seem a bit speculative in my view since neither the diffusion, nor any other MRI parameter could really specifically reflect this fine cellular arrangement. Maybe, since the paper already describes impressive meaningful results (Figs. 3-4) this section can be spared or moved to the supplementary information.

Reviewer #2 (Remarks to the Author):

The paper studies an important question, namely the relation between histological parameters from Nissl and Myelin stained images, and diffusion MRI data. The primary claim made in the study is that the fractional anisotropy in dMRI data correlates more strongly with a Nissl-derived anisotropy measure, rather than myelin content.

The study is interesting but has significant weaknesses.

1) There is no prima facie reason why the fractional anisotropy should correlate with myelin content per se, since the latter is a scalar quantity that could be high even if the myelinated fibers were not well orientated. In this case there would be no reason to expect a correlation between the diffusional anisotropy, fundamentally a tensor quantity, and myelin content, a scalar quantity.

To provide a fair comparison, the authors should also study the relation between the diffusional anisotropy, and Nissl content (defined in a scalar fashion like the Myelin content). In that case, one might also expect weak or no correlations. In fact, as the authors note, their considerations break down in layer 4 where there is a dense packing of granule cells.

The correlation between MTR and myelin content, is a clean result, but not the focus of the paper.

2) If one looks purely at the orientation of the diffusion anisotropy (Fig 7b, SI Fig 3) there seems to be good correspondence between the orientations visible in the Nissl-stained or Myelin-stained sections. The authors do not adequately emphasize or study this. If one were to look purely at orientation of the local anisotropy in diffusion, what does it correlate better with: the striations visible in the Nissl-stained sections (largely unmyelinated white matter), or the orientation of the Myelinated fibers?

Notably the Nissl stained somata may be organized in columnar structures, but this does not tell us what the unmyelinated axons are doing - these are invisible to the stain and there is no a priori reason to assume that the fibers are oriented parallel to the unstained channels around the somata.

3) There is no consideration of section thickness in the histology. 50 μ m sections are fairly thick on the scale of myelinated axons - especially in the myelin-rich regions, it is probably going to be quite difficult to resolve individual myelinated fibers at that thickness. The degree of anisotropy estimated from the myelinated images (which the authors do using structure tensor analysis), is going to be underestimated as a result. Thinner sections might produce more anisotropy in the structure tensor analysis of the myelin stained images. Admittedly these are challenging experiments to do, so the reviewer is not suggesting that the authors must go back and take thinner sections for this paper, but they should at least consider this caveat and carefully understand what this might imply for their analyses and conclusions.

In summary, while the paper is interesting, the authors seem to oversimplify and overstate their findings in order to make the outcome more striking - it would be better if they wrote a more nuanced paper with caveats better spelled out.

Reviewer #3 (Remarks to the Author):

There has been a surge in the interest in identifying the relationship between the results obtained with non-invasive (such as in-vivo MR imaging) and invasive (e.g. various histological stainings) experimental methods. Establishing these relations is essential in multiple contexts, from clinical, through experimental to computational studies.

A part of this debate is establishing which specific properties of the neural tissue drive the diffusion MR imaging (dMRI). In particular, how well the diffusion imaging reflects the actual structural (axonal,

dendritic) connectivity and the relation between these two approaches.

The main focus has been on the white matter, where assessing such relations seems less challenging. This study focuses on the cortical grey matter to identify which properties of neuronal tissue drive the dMRI imaging.

From this perspective, the article is interesting, valuable and timely.

The main finding is that the dMRI fractional anisotropy (dMRI-FA) correlates well with the histological anisotropy (HA, an index introduced by the authors to characterize a predominant orientation of the histological and cytoarchitectural features) rather than the amount of myelinated features or their primary orientation. The authors also report that the myelin content represented is far better by the magnetization transfer ratio (MTR).

Authors conclude that the dMRI-FA in the cerebral cortex grey matter is determined mainly by unmyelinated tissue components such as dendrites, glial processes and unmyelinated axons.

From the experimental perspective, the study relies on MR scans of various modalities and histological sections (thionin and Gallyas stains) from a single marmoset brain. Both the MR imaging and histological procedures are adequate from the perspective of the follow-up computational analysis.

The computational framework is conceptually straightforward and relies on comparing the properties of the diffusion tensor model of the dMRI signal with its histological counterpart (Nissl-HA or myelin-HA).

MAJOR REMARKS

A major concern is that the results of the analyses support the author's hypothesis only indirectly by eliminating other likely factors rather than presenting direct evidence. Therefore I would consider the presented arguments insufficient. To this end:

107-109: (...) dMRI-signal in the cerebral cortex is determined principally by unmyelinated components of the tissue, such as dendrites, glial processes and unmyelinated axons.

The Nissl (or myelin) -HA measures are derived from the images of the stained sections. However, the unmyelinated components are not visible in either stain. Therefore the collected experimental material does not allow one to draw the conclusions presented by the authors directly. This indirect nature of the Nissl-HA is highlighted in the discussion (L288-290) by the authors themselves.

The titular statement could be supported more directly, for instance, by using more comprehensive range of staining techniques. To this end, the authors mention (L312-318) that there are readily available experimental techniques (which are also relatively accessible) that would make it possible to unambiguously and directly verify the author's hypothesis and to strengthen the conclusions.

In addition, throughout the article, I could not find any information on which (anatomically defined) cortical areas the analysis covered. Authors use the term "patch", which carry no anatomical meaning. Further, a single patch is likely to cover many cortical areas (e.g. adjacent agranular and granular areas) therefore blending the cytoarchitectural features of the tissue. For instance, Figure 1a depicts a part of the cortex that stretches from V1 to the prefrontal cortex covering areas of various cyto- and myelinoarchitecture.

The study could become much more insightful if the analysis could also consider the division of the tissue into cortical areas. In the worst-case scenario, it would show that the results are consistent

regardless of the area considered. However, such analysis would likely reveal more interesting details and relations similar to those in layer 4 (Fig. 7A, L249-253).

MINOR REMARKS

L136-137: "(...) there are areas in which they appear to match (blue arrow) and others in which they clearly do not (red arrow)"

It seems that no attempt was made to identify the reason for such a (miss)match. Perhaps this is related to some specific features of the examined cytoarchitectural areas?

Figure 3, bottom row: The scatterplots (which, I assume, represent individual pixels of an example "patch") exhibit a more complex structure than merely a high correlation or a lack of thereof. The dMRI-FA vs. myelin intensity presents a relation in which the dMRI-FA is first proportional to the myelin intensity only to become inversely proportional for high myelin intensity. In addition, in the dMRI-FA vs. Nissl-HA relation, there actually are two bands of points. Have the authors attempted to examine these complex relations? Could they be attributed to any particular part of a given patch (e.g. specific layer)? Do these relations appear only in this patch or are they representative to all patches, hence all examined cortical areas? This question becomes even more appropriate in the context of the section "The exception of granular layer 4".

Figure 4, the title of the plot on panel B: it should probably state "MTR" instead of "dMRI-FA".

Figure 5, panel A - "Normalized magnitude" - the scale is missing.

SI Figure 5. It should be clarified what the term "size" means. From the context I would assume that it is the length of the segment of the pial surface based on which the cortical "parcel" is defined. However, this seems not to be clarified in the text.

Figure 7, panel B: the figure would be much more legible if the borders between the individual layers could be marked on the profiles (both, the Nissl and the myelin). Also the 3D model could be aligned so that the borders on in the 3D model match respective borders annotated on the profiles.

Overall, in its current form, the article is a solid and relatively thorough study on the relation between dMRI-derived measures and their cytoarchitectural counterparts which is valuable by itself. However, the main thesis of the article is supported only indirectly and more direct evidence and analyses are required to address the titular thesis unambiguously.

Further, there are a few unexplored possibilities that definitely should be addressed. Most of them require including the parcellation into cortical areas in the analyses.

Reviewer #1 (Remarks to the Author):

The paper by Reveley et al explore the cellular relevance of diffusion MRI measurements in the cortex.

Before providing my review of this paper, I would like to note that over the development of diffusion MRI as a tool in neuroscience it became fixated that this method is used for white matter mapping (mainly due to fiber-tracking and its consequent mapping of white matter structure in 3D). I strongly believe, as the paper points out, that diffusion in the cortex is highly meaningful and provide micro-structural view of cortical architecture. For that reason, and due to the popularity of diffusion MRI in neuroscience and it's unfortunate tagging as white matter mapping tool – I think this paper deserves high-impact publication.

The paper nicely demonstrates through direct comparison between one Marmoset brain MRI (diffusion and MTR) with various histological measures the relation between anisotropy measurements in the cortex and cellular morphology.

Specifically, I think Figures 3 and 4 demonstrates that main result of this paper (mainly Fig. 3 dMRI-FA/Nissl-HA and MTR/Myelin intensity graphs; please consider adding A, B, C, D tagging to this figure) and those should be highlighted over other results.

We thank the reviewer for these comments, which indeed reflect the overarching motivation for our research in this paper. As suggested by the reviewer, we have now added A,B tagging to figure 3.

I have only minor comments for the authors to consider:

1. Abstract: "Those unmyelinated features" needs better phrasing - Dendrites? Neural Processes? Unmyelinated Axons? (all of them?). Need to be more explicit.

We appreciate this opinion, which was shared by other reviewers. Based on further analysis, we now highlight the putative contribution of apical dendrites in shaping the FA. We discuss this in the paper (adding a new Figure 6) and have added it to the abstract. In the Discussion, we draw upon several studies that link the presence of such dendrites to the observed Nissl anisotropy, and discuss how this observation helps to understand the observed mismatch between FA and Nissl anisotropy in layer 4 of highly granular cortex.

2. Introduction, general – The authors state that diffusion MRI in the cortex is dominated by fiber arrangement perpendicular to pial surface. This is evidently true. However, in such a heterogeneous environment - does this perpendicular dominance really significant? In my view - only more generalized models than the diffusion tensor (e.g. spherical harmonics) can capture the complexity of organization in the cortex (some of the references the author cite also indicates that). Yet, the validity of the simple diffusion tensor model increases with extreme resolution where partial volume induced complexity reduces dramatically. It appears that this is the case in this work – 150micron resolution in the cortex should be sufficient so that the diffusion tensor model can be used. Maybe the authors would benefit from acknowledging that diffusion in the cortex necessitates more complicated models, but due to high resolution that is used, the simple diffusion tensor is adequate.

We appreciate this comment and have now made reference to more complex cortical models. We have also altered the text to make clear that the principal orientation in the cortex is always vertical, whether measured using histological stains or diffusion MRI methods.

3. Introduction, last paragraph - Histological anisotropy - is that a common index of histology? A reference would be helpful. Later on in the results another terminology is used - structural tensor (ST) - this reminds me of a procedure by Budde that also explored correlation between histology and dMRI in the cortex. Is that a similar procedure? If so, a reference could be helpful.

After reading the methods section as well, I realized that indeed this is the case – so just reference Budde also in the main text.

We thank the reviewer for this reminder. Our measure is a slightly augmented structure tensor, and we now reference Budde and Frank's 2012 paper in the introduction.

4. 58 hours of scanning – Was the data corrected for signal drift? From my experience such long scanning may suffer from drift artifact which may have devastating effect on the diffusion anisotropy measurements.

We appreciate this comment. It is indeed true that large frequency drift can take place in scans lasting as long as ours did. However, such drift could not cause problems in our data because they were not acquired in interleaved fashion within the 58 hour period. Briefly, our data consist of 124 volumes of 3D brain images. The volumes were acquired sequentially, with each volume acquisition taking about 28 minutes. The frequency drift in the scanner over 28 minutes acquisition was not large enough to seriously affect the EPI quality. Thus, the effect of frequency drift between volumes, if present, was manifest as an image shift in the phase-encoding direction, which was readily corrected using standard image registration. Further, to overcome potentially large frequency shifts accruing over the 58 hours, the scans were divided into 12 batches, with the scanner set to recalibrate its resonance frequency between the batches. In other words, frequency drift was never allowed to develop longer than 6 hours.

5. What does the blue circles in the violin plot (Fig 4) indicates?

We have altered figure 4 and the explanatory text in a way that now explicitly indicates the meaning of the circles, which are the correlation coefficients for each parcel of cortex. Based on a suggestion from another reviewer, they are now coloured by the anatomical region into which they fall.

6. The Paper is long. The main message is already well received after Fig.4. The RD/AD correlation could be moved to the supplementary material. The layer 4 neuronal type analysis (Fig 6 and on) seem a bit speculative in my view since neither the diffusion, nor any other MRI parameter could really specifically reflect this fine cellular arrangement. Maybe, since the paper already describes impressive

meaningful results (Figs. 3-4) this section can be spared or moved to the supplementary information.

We appreciate this comment we agree with the general sentiment. To address this, we have now moved figure 6 and 7 from the previous draft into supplemental material. We have kept an abridged discussion of diffusivity in the main text, as we feel it is important to mention because it highlights the distinct mechanisms by which myelinated and non-myelinated tissue features affect anisotropy.

We appreciate the reviewer's thoughtful and constructive comments, which we believe have resulted in a clearer and better structured paper.

Reviewer #2 (Remarks to the Author):

The paper studies an important question, namely the relation between histological parameters from Nissl and Myelin stained images, and diffusion MRI data. The primary claim made in the study is that the fractional anisotropy in dMRI data correlates more strongly with a Nissl-derived anisotropy measure, rather than myelin content.

We thank the reviewer for these remarks regarding the importance of our research question.

1) There is no prima facie reason why the fractional anisotropy should correlate with myelin content per se, since the latter is a scalar quantity that could be high even if the myelinated fibers were not well orientated. In this case there would be no reason to expect a correlation between the diffusional anisotropy, fundamentally a tensor quantity, and myelin content, a scalar quantity.

We appreciate this point, which we agree with wholeheartedly. Indeed, much of the paper is devoted to the structure tensor analysis of the histological data rather than to its scalar aspect (e.g. myelin intensity). A central finding of the study is that neither the intensity nor the structure tensor derived from myelin staining matches the observed diffusion anisotropy.

That said, we do feel it is warranted to present the study in the context of myelin intensity, since the literature does frequently associate myelin intensity with FA¹⁻³⁴. It is also worth mentioning that the principal tensor direction derived from diffusion matches the orientation of myelin very well (Fig 2, SI Fig 4). However, in the cerebral cortex, we show that the spatial *variation* of diffusion anisotropy does not match that of myelin, either in its intensity or local orientation anisotropy. We have altered the language throughout the paper to make these points clearer.

To provide a fair comparison, the authors should also study the relation between the diffusional anisotropy, and Nissl content (defined in a scalar fashion like the Myelin content). In that case, one might also expect weak or no correlations. In fact, as the authors note, their considerations break down in layer 4 where there is a dense packing of granule cells.

Based on this comment, we performed this analysis of Nissl content, now shown in SI Fig 2. Briefly, we found that Nissl content had a poor correlation with diffusion FA in the cerebral cortex. We have now added a sentence to this effect in the Results.

2) If one looks purely at the orientation of the diffusion anisotropy (Fig 7b, SI Fig 3) there seems to be good correspondence between the orientations visible in the Nissl-stained or Myelin-stained sections. The authors do not adequately emphasize or study this. If one were to look purely at orientation of the local anisotropy in diffusion, what does it correlate better with: the striations visible in the Nissl-stained sections (largely unmyelinated white matter), or the orientation of the Myelinated fibers?

We thank the reviewer for this comment. Indeed, the principal orientations are always vertical, in the Nissl sections (as the reviewer points out), the myelin sections and the diffusion MRI tensors. We demonstrate this in Fig 2 and SI Fig 4. We found that the mean angles of the principal orientation were very closely aligned in all data modalities, reflecting the basic structural organization of the cortex. We added text to distinguish this point from the main task of identifying the histological basis of spatial *variation* in diffusion anisotropy in the cerebral cortex.

Notably the Nissl stained somata may be organized in columnar structures, but this does not tell us what the unmyelinated axons are doing - these are invisible to the stain and there is no a priori reason to assume that the fibers are oriented parallel to the unstained channels around the somata.

We appreciate and agree with this comment. We know from the literature that both unmyelinated axons and other unmyelinated neurites, such as apical dendrites, tend to form bundles as they pass vertically through the cortex. We believe the apical dendrites may be of particular importance and have added sentences to that effect in the revision. To illustrate their structure in the marmoset, we have now added a MAP-2 stained section of the cortex of a second animal. MAP-2 stains cell bodies and large caliber dendrites and highlights what we believe is the most prominent contributor to the observed match between the Nissl anisotropy and diffusion anisotropy. In the text, we now link our analysis of Nissl data more closely to the literature of the columnar neurite architecture of the cortex, in which many authors have found that columns of cell bodies in the cortex do indeed reflect the organisation of non-myelinated neurites e.g. ⁵⁻⁹, especially apical dendrites.

There is no consideration of section thickness in the histology. 50µm sections are fairly thick on the scale of myelinated axons - especially in the myelin-rich regions, it is probably going to be quite difficult to resolve individual myelinated fibers at that thickness. The degree of anisotropy estimated from the myelinated images (which the authors do using structure tensor analysis), is going to be underestimated as a result. Thinner sections might produce more anisotropy in the structure tensor analysis of the myelin stained images. Admittedly these are challenging experiments to do, so the reviewer is not suggesting that the authors must go back and take thinner sections for this paper, but they should at least consider this caveat and carefully understand what this might imply for their analyses and conclusions.

We appreciate these points and have now added two supplemental figures to address this concern. In SI Fig 16, we demonstrate that we were able to distinguish individual axons in the myelin rich regions. Substantial robustness to variation in stain intensity and/or section thickness is one of the advantages of the ST coherence measure. This is because the measure is sensitive to the distribution of angles in the image edges, not the density of edges (see **Methods**). In SI Fig 17, we confirm this robustness empirically by resolving fibers at different depths of the 50um section by varying the focal plane. Resolving the section at different depths had minimal effect on ST coherence as the orientation distributions of the near and far fibers were similar.

We appreciate the reviewer's careful consideration of the data in the paper and many constructive comments. In following each of the suggestions, we believe that the paper is significantly strengthened.

Reviewer #3 (Remarks to the Author):

There has been a surge in the interest in identifying the relationship between the results obtained with non-invasive (such as in-vivo MR imaging) and invasive (e.g. various histological stainings) experimental methods. Establishing these relations is essential in multiple contexts, from clinical, through experimental to computational studies.

A part of this debate is establishing which specific properties of the neural tissue drive the diffusion MR imaging (dMRI). In particular, how well the diffusion imaging reflects the actual structural (axonal, dendritic) connectivity and the relation between these two approaches.

The main focus has been on the white matter, where assessing such relations seems less challenging. This study focuses on the cortical grey matter to identify which properties of neuronal tissue drive the dMRI imaging.

From this perspective, the article is interesting, valuable and timely.

The main finding is that the dMRI fractional anisotropy (dMRI-FA) correlates well with the histological anisotropy (HA, an index introduced by the authors to characterize a predominant orientation of the histological and cytoarchitectural features) rather than the amount of myelinated features or their primary orientation. The authors also report that the myelin content represented is far better by the magnetization transfer ratio (MTR).

Authors conclude that the dMRI-FA in the cerebral cortex grey matter is determined mainly by unmyelinated tissue components such as dendrites, glial processes and unmyelinated axons.

From the experimental perspective, the study relies on MR scans of various modalities and histological sections (thionin and Gallyas stains) from a single

marmoset brain. Both the MR imaging and histological procedures are adequate from the perspective of the follow-up computational analysis.

The computational framework is conceptually straightforward and relies on comparing the properties of the diffusion tensor model of the dMRI signal with its histological counterpart (Nissl-HA or myelin-HA).

We thank the reviewer for this positive assessment of our research question and general approach.

MAJOR REMARKS

A major concern is that the results of the analyses support the author's hypothesis only indirectly by eliminating other likely factors rather than presenting direct evidence. Therefore, I would consider the presented arguments insufficient. To this end:

107-109: (...) dMRI-signal in the cerebral cortex is determined principally by unmyelinated components of the tissue, such as dendrites, glial processes and unmyelinated axons.

The Nissl (or myelin) -HA measures are derived from the images of the stained sections. However, the unmyelinated components are not visible in either stain. Therefore, the collected experimental material does not allow one to draw the conclusions presented by the authors directly. This indirect nature of the Nissl-HA is highlighted in the discussion (L288-290) by the authors themselves.

We thank the reviewer for raising this concern. We have now rewritten much of the text in the manuscript. We now show that the link between the cellular columns shown in Nissl data, and non-myelinated neurites, especially apical dendrite bundles, is strong in the literature. We mention several studies that have explored this, for example ⁵⁻⁹. We also highlight this link in the marmoset using immunohistochemistry in a second subject and presented the data in a new figure (**Figure 6**).

The titular statement could be supported more directly, for instance, by using more comprehensive range of staining techniques. To this end, the authors mention (L312-318) that there are readily available experimental techniques (which are also relatively accessible) that would make it possible to unambiguously and directly verify the author's hypothesis and to strengthen the conclusions.

We thank the reviewer for making this point. The term "unmyelinated" in the title reflects the fact that the myelin stain does, in fact, isolate one specific tissue feature, namely myelinated axons. Rightly or wrongly, there is a significant association in the literature between FA and myelin, including in the cerebral cortex ¹⁻⁴. In our view, the analysis applied to myelin-stained tissue is sufficient to support the term "unmyelinated" in the title.

Going further to state what the specific unmyelinated components are is, naturally, more difficult. We appreciate that the reviewer has pressed us on this issue, which has led us to emphasize more strongly the feature that we believe plays the strongest role, namely the apical dendrites. In the revised text, we link the pattern of Nissl staining to apical dendrite structure, drawing upon literature that has previously made this connection⁵⁻⁹. We also added a new figure (Fig 6) highlighting the apical dendrite structure in the marmoset cortex revealed by MAP-2 staining. While the specific contribute of apical dendrite and other features requires further study, we can state unequivocally that the important contributors are unmyelinated.

In addition, throughout the article, I could not find any information on which (anatomically defined) cortical areas the analysis covered. Authors use the term “patch”, which carry no anatomical meaning. Further, a single patch is likely to cover many cortical areas (e.g. adjacent agranular and granular areas) therefore blending the cytoarchitectural features of the tissue. For instance, Figure 1a depicts a part of the cortex that stretches from V1 to the prefrontal cortex covering areas of various cyto- and myelinoarchitecture. The study could become much more insightful if the analysis could also consider the division of the tissue into cortical areas. In the worst-case scenario, it would show that the results are consistent regardless of the area considered. However, such analysis would likely reveal more interesting details and relations similar to those in layer 4 (Fig. 7A, L249-253).

Based on the reviewer’s suggestions, we have now incorporated information about anatomical areas in the main analysis. Specifically, Fig 4 now indicates the cortical area, derived from the Paxinos¹⁰ atlas, corresponding to each of the parcels for which the diffusion data was correlated with the histological data. The basic findings are robust to changes in the parcel size (SI Fig 7), including for small parcels for which there is minimal problem with individual parcels spanning areal boundaries. In general, the observed correlation structure did not vary systematically across cortical areas, with the exception that agranular regions tended to exhibit the strongest correlation between Nissl-HA and FA.

MINOR REMARKS

L136-137: “(...) there are areas in which they appear to match (blue arrow) and others in which they clearly do not (red arrow)”

It seems that no attempt was made to identify the reason for such a (miss)match. Perhaps this is related to some specific features of the examined cytoarchitectural areas?

We agree that we did not speculate on the reason for the basis for matching or mismatching in this particular example. Our purpose with this early figure was mainly to show the unreliability of myelin as a marker for FA. Even when it might seem at lower resolution that the horizontal variation in FA matches that of myelin, in high resolution data the laminar distribution does not match. A more comprehensive consideration of the mismatch of dMRI-FA with myelin comes later in the paper.

Figure 3, bottom row: The scatterplots (which, I assume, represent individual pixels of an example “patch”) exhibit a more complex structure than merely a high

correlation or a lack of thereof. The dMRI-FA vs. myelin intensity presents a relation in which the dMRI-FA is first proportional to the myelin intensity only to become inversely proportional for high myelin intensity. In addition, in the dMRI-FA vs Nissl-HA relation, there actually are two bands of points. Have the authors attempted to examine these complex relations? Could they be attributed to any particular part of a given patch (e.g. specific layer)? Do these relations appear only in this patch or are they representative to all patches, hence all examined cortical areas? This question becomes even more appropriate in the context of the section “The exception of granular layer 4”.

We thank the reviewer this comment. In order to examine these relationships in more detail, we computed the average laminar variation of myelin intensity (**Figure 1b**), dMRI-FA (**Figure 1b, Figure 5a**), Nissl-HA and myelin-HA (**Figure 5a**). As expected from the literature, these gray matter features follow gradients across a portion of the cortical thickness. These partial gradients lead to the observed piecewise correlations. For example, in Figure 3 the scatterplot tracks the monotonic decline for myelin intensity as it relates to a more parabolic laminar trend in the dMRI-FA. This is a central factor affecting the main correlational analysis.

Figure 4, the title of the plot on panel B: it should probably state “MTR” instead of “dMRI-FA”.

We thank the reviewer for this correction.

Figure 5, panel A - “Normalized magnitude” - the scale is missing.

We thank the reviewer for this correction.

SI Figure 5. It should be clarified what the term “size” means. From the context I would assume that it is the length of the segment of the pial surface based on which the cortical “parcel” is defined. However, this seems not to be clarified in the text.

We have added this definition to the figure legend (now **SI Figure 7**). It is also present in the text.

Figure 7, panel B: the figure would be much more legible if the borders between the individual layers could be marked on the profiles (both, the Nissl and the myelin). Also, the 3D model could be aligned so that the borders on in the 3D model match respective borders annotated on the profiles.

We thank the reviewer for this correction.

1. Lazari, A. & Lipp, I. Can MRI measure myelin? Systematic review, qualitative assessment, and meta-analysis of studies validating microstructural imaging with myelin histology. *Neuroimage* **230**, 117744 (2021).
2. Van der Weijden, C. W. J. *et al.* Myelin quantification with MRI: A systematic review of accuracy and reproducibility. *Neuroimage* **226**, 117561 (2020).
3. Mancini, M. *et al.* An interactive meta-analysis of MRI biomarkers of Myelin.

- Elife* **9**, 1–23 (2020).
4. Howard, A. F. D. *et al.* Estimating intra-axonal axial diffusivity with diffusion MRI in the presence of fibre orientation dispersion. *bioRxiv* 1–33 (2020).
 5. Gabbott, P. L. A. Radial organisation of neurons and dendrites in human cortical areas 25, 32, and 32'. *Brain Res.* **992**, 298–304 (2003).
 6. Gabbott, P. & Bacon, S. . The organisation of dendritic bundles in the prelimbic cortex (area 32) of the rat. *Brain Res.* **730**, 75–86 (1996).
 7. Buxhoeveden, D. P. & Casanova, M. F. The minicolumn hypothesis in neuroscience. *Brain* **125**, 935–951 (2002).
 8. Buldyrev, S. V. *et al.* Description of microcolumnar ensembles in association cortex and their disruption in Alzheimer and Lewy body dementias. *Proc. Natl. Acad. Sci. U. S. A.* **97**, 5039–5043 (2000).
 9. Peters, A. The Morphology of Minicolumns. *Neurochem. Basis Autism From Mol. to Minicolumns* 1–295 (2010). doi:10.1007/978-1-4419-1272-5
 10. Paxinos, G., Watson, C., Petrides, M., Rosa, M. & Tokuno, H. *The Marmoset Brain in Stereotaxic Coordinates.* (2013).

REVIEWER COMMENTS

Reviewer #1 (Remarks to the Author):

None

Reviewer #2 (Remarks to the Author):

The authors have addressed most of the concerns raised in the previous review with additional analyses, paper re-writing, and even adding new data. Only two minor comments at this point:

(1) This sentence "We found that cortical diffusion was only minimally related to the density and arrangement of myelinated fibers. " in the abstract is an oversimplification and does some injustice to the more nuanced and careful analysis presented in the manuscript. While the degree of anisotropy in the MRI diffusion tensor does not relate to myelin density (the main point of the paper, and a good one to emphasize), the authors nevertheless show that there is indeed some correlation between the diffusional fractional anisotropy and the myelin histological anisotropy, though the correlation is stronger with the nissl histological anisotropy (and more region specific). Moreover, the *orientation* of the largest eigenvector of the diffusion tensor as well as the nissl and myelin anisotropies are all aligned (and perpendicular to the cortical axis). Thus it is not quite correct to say that the cortical diffusion is minimally related to the arrangement of the myelinated fibers - the radial direction shows up in all the measures including the arrangement of the myelinated fibers.

This is a minor quibble but the reviewer recommends that the authors nuance their abstract in the same way they have nuanced the paper discussion - often the abstract is what many readers will carry away from the manuscript so it is worthwhile for the authors to get the language quite right.

(2) The strongest correlation shown in the paper is between myelin density and (ex-vivo) MTR. This high degree of correlation is remarkable and may have utility in biomarker development; one natural question (although the reviewer does not expect the reviewers to do more experiments) is whether this is also expected in-vivo. If the authors have any comments on this matter they should include it in the discussion.

Reviewer #3 (Remarks to the Author):

My major remarks concerned the indirect nature of the Nissl-HA as an indicator of the unmyelinated features. The recommended remedy was to strengthen the author's argument with more direct evidence. The authors took these remarks seriously and expanded the manuscript in the recommended direction by 1) adding Figure 6. and 2) by identifying the apical dendrites as a major contributor to the observed phenomenon.

The only (optional) change I would suggest would be to extend Fig. 6 with a corresponding "patch" showing myelin stain of that part of the cortex.

The other major remark was the lack of association between the analyzed "patches" and their relation to the anatomically defined cortical areas. I believe this issue has been also comprehensively addressed by extending Fig. 4. The high correlation between Nissl-HA and the agranular regions seems to be logical and to follow other observations (SI Figure 3).

All the minor remarks have also been addressed.

We thank all three reviewers for their very helpful and constructive comments, and their assistance in tightening our manuscript.

REVIEWERS' COMMENTS

Reviewer #1 (Remarks to the Author):

None

Reviewer #2 (Remarks to the Author):

The authors have addressed most of the concerns raised in the previous review with additional analyses, paper re-writing, and even adding new data. Only two minor comments at this point:

(1) This sentence "We found that cortical diffusion was only minimally related to the density and arrangement of myelinated fibers. " in the abstract is an oversimplification and does some injustice to the more nuanced and careful analysis presented in the manuscript. While the degree of anisotropy in the MRI diffusion tensor does not relate to myelin density (the main point of the paper, and a good one to emphasize), the authors nevertheless show that there is indeed some correlation between the diffusional fractional anisotropy and the myelin histological anisotropy, though the correlation is stronger with the nissl histological anisotropy (and more region specific). Moreover, the *orientation* of the largest eigenvector of the diffusion tensor as well as the nissl and myelin anisotropies are all aligned (and perpendicular to the cortical axis). Thus it is not quite correct to say that the cortical diffusion is minimally related to the arrangement of the myelinated fibers - the radial direction shows up in all the measures including the arrangement of the myelinated fibers.

This is a minor quibble but the reviewer recommends that the authors nuance their abstract in the same way they have nuanced the paper discussion - often the abstract is what many readers will carry away from the manuscript so it is worthwhile for the authors to get the language quite right.

We thank the reviewer for this comment. We've now adjusted the language in abstract.

(2) The strongest correlation shown in the paper is between myelin density and (ex-vivo) MTR. This high degree of correlation is remarkable and may have utility in biomarker development; one natural question (although the reviewer does not expect the reviewers to do more experiments) is whether this is also expected in-vivo. If the authors have any comments on this matter they should include it in the discussion.

We thank the reviewer for this comment. We did not include in-vivo data in this study and prefer not to speculate.

Reviewer #3 (Remarks to the Author):

My major remarks concerned the indirect nature of the Nissl-HA as an indicator of the unmyelinated features. The recommended remedy was to strengthen the author's argument with more direct evidence. The authors took these remarks seriously and expanded the

manuscript in the recommended direction by 1) adding Figure 6. and 2) by identifying the apical dendrites as a major contributor to the observed phenomenon.

The only (optional) change I would suggest would be to extend Fig. 6 with a corresponding "patch" showing myelin stain of that part of the cortex.

We thank the reviewer for this suggestion. Myelin histology was unavailable in the subject for which MAP-2 immunohistochemistry was obtained.

The other major remark was the lack of association between the analyzed "patches" and their relation to the anatomically defined cortical areas. I believe this issue has been also comprehensively addressed by extending Fig. 4. The high correlation between Nissl-HA and the agranular regions seems to be logical and to follow other observations (SI Figure 3).

All the minor remarks have also been addressed.